# Geometric and electronic structure probed along the isomerisation coordinate of a photoactive yellow protein chromophore

Cate S. Anstöter [1], Basile F. E. Curchod [1] & Jan R. R. Verlet [1✉]

Understanding the connection between the motion of the nuclei in a molecule and the rearrangement of its electrons lies at the heart of chemistry. While many experimental methods have been developed to probe either the electronic or the nuclear structure on the timescale of atomic motion, very few have been able to capture both these changes in concert. Here, we use time-resolved photoelectron imaging to probe the isomerisation coordinate on the excited state of an isolated model chromophore anion of the photoactive yellow protein. By probing both the electronic structure changes as well as nuclear dynamics, we are able to uniquely measure isomerisation about a specific bond. Our results demonstrate that the photoelectron signal dispersed in time, energy and angle combined with calculations can track the evolution of both electronic and geometric structure along the adiabatic state, which in turn defines that chemical transformation.

[1] Department of Chemistry, Durham University, Durham DH1 3LE, UK. ✉email: j.r.r.verlet@durham.ac.uk

Chemistry can be broadly defined as the change in valence electronic structure as atoms in a molecule geometrically rearrange. The adiabatic picture that describes this delicate interplay between electrons and nuclei is a central pillar to chemical dynamics and leads to the concept of a potential energy surface within the Born-Oppenheimer approximation[1]. Consequently, developing experimental probes that are sensitive to both nuclear and electronic evolution in real-time has become a primary goal in chemical physics[2–6]. Time-resolved photoelectron spectroscopy is one such method as it maps an adiabatically evolving wavepacket onto ionised final states with no strict selection rules[7,8]. A variant of the method, time-resolved photoelectron imaging, offers an additional dimension through the photoelectron angular distributions (PADs) that are sensitive to the molecular orbital from which the electron is removed[9]. Adiabatic changes can be tracked through molecular-frame PADs, but such measurements require a connection between the laboratory and molecular frames of reference, either through coincidence measurements[2,5] or molecular alignment[10]. This complexity inhibits the application of such experiments to complex polyatomic molecules for which the methods are ultimately designed. Overcoming these limitations will provide a platform to probe chemical dynamics in complex molecules.

The photoactive yellow protein, PYP, is a blue-light absorbing protein that has been an important testbed for novel structural probes in complex biological environments[11,12]. The absorption maximum for the $S_1 \leftarrow S_0$ transition in PYP is at ~450 nm and can be traced to a small chromophore that undergoes a light-activated *trans–cis* isomerisation which serves as a mechanical lever and triggers an extensive bio-cycle with numerous intermediates[13–15]. Derivatives of the PYP chromophore are commonly based on *para*-coumaric acid and have been studied extensively as a prototypical bio-chromophore[16,17]. Yet, there remains ambiguity about which specific bonds are involved in the initial excited state isomerisation and, hence, there is a desire to develop experimental probes that can distinguish subtly different reaction coordinates. For example, the anionic *para*-coumaric ketone chromophore ($p$CK$^-$, Fig. 1a), studied by Zewail and coworkers using time-resolved photoelectron spectroscopy, can isomerise about the first (single), the second (double), or both bonds in the *para*-position; but the photoelectron spectra alone could not discern these differences[16]. Chemical derivatives in which rotation about specific bonds is inhibited have also been studied, but such modifications diverge further from the native chromophore[18]. Several computational studies have explored the potential energy surfaces of the $S_0$ and $S_1$ states and considered the dynamics on the $S_1$ state following photoexcitation[19–23]. These have converged on the position that the initial isomerisation coordinate involves predominantly rotation about the single bond, but have not been clearly linked with experimental data. In the present study, $p$CK$^-$ is probed by time-resolved photoelectron imaging combined with electronic structure calculations. For specific photoelectron features, the temporal evolution of the spectra and laboratory-frame PADs, in unison with our calculations, enables the identification of the nuclear and electronic structural changes associated with the single-bond isomerisation coordinate on the excited state, thus demonstrating a direct probe for adiabatic dynamics, i.e. chemistry.

## Results

### Time-resolved photoelectron imaging.
Our experiment involves excitation of mass-selected $p$CK$^-$ at 2.79 eV (444 nm) with femtosecond pulses to the bright $S_1$ state. The excited state dynamics are subsequently probed at various delays using femtosecond pulses at 1.55 eV (800 nm) at the centre of a velocity

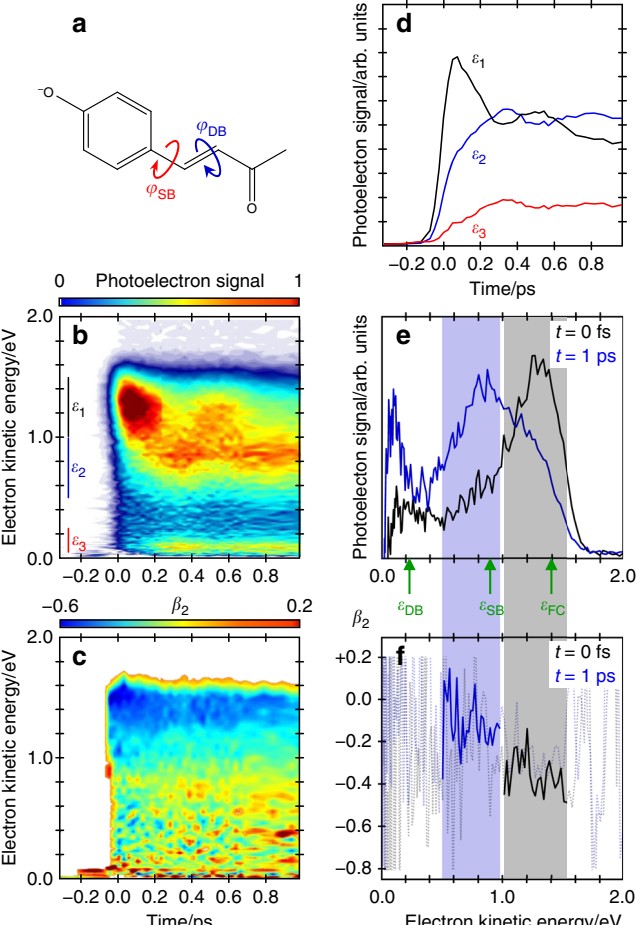

**Fig. 1 Excited state dynamics of $p$CK$^-$. a** Structure of $p$CK$^-$ with arrows indicating single ($\varphi_{SB}$) and double ($\varphi_{DB}$) bond rotation. **b** Time-resolved photoelectron spectra plotted as a normalised false-colour map with a $t < 0$ spectrum subtracted and where white indicates signal <0. **c** time-resolved $\beta_2$ false-colour map, where values are omitted whenever the signal level is <0.1 in **b**. **d** Integrated photoelectron signal over spectral windows $\varepsilon_1$, $\varepsilon_2$, and $\varepsilon_3$ indicated in **b**. **e** Photoelectron spectra at $t = 0$ and 1 ps, representative of spectral windows $\varepsilon_1$ and $\varepsilon_2$ in **b**, respectively. **f** $\beta_2$ corresponding to spectra in **e** as indicated by shaded regions. Vertical arrows in **d** show the maximum kinetic energy expected for detachment from the Franck-Condon, single-bond, and double bond rotated minima, $\varepsilon_{FC}$, $\varepsilon_{SB}$, and $\varepsilon_{DB}$, respectively.

map imaging spectrometer, yielding time-resolved photoelectron images. The temporal resolution (full width at half maximum) is $100 \pm 10$ fs and the spectral resolution ~5% of the electron kinetic energy, $\varepsilon$.

The time-resolved photoelectron spectra are shown in Fig. 1b over the first picosecond following excitation. Each spectrum has had the spectrum at $t < 0$ removed to leave only the time-evolving signals. Figure 1e displays spectra at a few specific delays. At very early times, the photoelectron spectrum exhibits a peak centred at an electron kinetic energy, $\varepsilon \sim 1.4$ eV. With increasing pump-probe delay, $t$, this initial peak shifts towards lower $\varepsilon$, leaving a peak centred at $\varepsilon \sim 0.8$ eV at longer times ($t \gg 1$ ps). This peak then slowly decays with a lifetime of ~120 ps with no further spectral evolution (see Supplementary Fig. 1). Based on our photoelectron spectra (Supplementary Note 2 and Supplementary Fig. 2), the electron affinity of $p$CK is $2.87 \pm 0.05$ eV. Hence, excitation to the $S_1$ state at 2.79 eV is just below the detachment threshold and ensures we are predominantly probing bound-state

dynamics, although the $S_1$ absorption profile is broad and extends into the continuum[24].

Fig. 1d presents the integrated photoelectron signal over specific spectral windows that are indicated in Fig. 1b ($\varepsilon_1$, $\varepsilon_2$, and $\varepsilon_3$) and are representative of the different spectral features. The high energy peak (represented by spectral region $\varepsilon_1$ in Fig. 1b) shows a rapid initial decay with an apparent oscillation super-imposed. The signal in the intermediate spectral range ($\varepsilon_2$) rises as the high energy signal decays and similarly oscillates with a commensurate period, but with a $\pi$ phase shift. These dynamics are also clearly visible in Fig. 1b.

In addition to the evolution of the photoelectron spectra, we also observe an evolution of the PADs. Laboratory-frame PADs are typically quantified by anisotropy parameters, $\beta_{2n}(\varepsilon)$[25]. For a two-photon (pump-probe) process, $n = 1$ and 2, and the PADs are defined through[9,26]

$$I(\varepsilon, \theta) = \sigma/4\pi\left[1 + \beta_2(\varepsilon)P_2(\cos\theta) + \beta_4(\varepsilon)P_4(\cos\theta)\right], \quad (1)$$

where $I(\varepsilon, \theta)$ is the photoelectron signal as a function of the angle, $\theta$, between the laser polarisation axis and the photoelectron emission velocity vector, $\sigma$ is the detachment cross-section, and $P_2(\cos\theta)$ and $P_4(\cos\theta)$ are the second- and fourth-order Legendre polynomials. For large polyatomic molecules, only changes in $\beta_2$ are often significant, which has limiting values $+2$ and $-1$ that correspond to electron emission predominantly parallel to and perpendicular to the polarisation axis, respectively. Figure 1c shows the measured $\beta_2(\varepsilon, t)$, with a 5-point moving average in $\varepsilon$. Figure 1c is directly comparable to the spectral evolution shown in Fig. 1b. Note that when the overall photoelectron signal is low, the determination of $\beta_2(\varepsilon, t)$ has a large uncertainty and we omit data for signal that is less than 0.1 of the normalised signal in Fig. 1b for clarity. The corresponding $\beta_4(\varepsilon, t)$ data are given in the Supplementary Note 3 and Supplementary Fig. 3 and have values very close to zero suggesting that $\beta_2(\varepsilon, t)$ is a good measure of the overall PADs.

Figure 1f shows the $\beta_2(\varepsilon)$ with no moving-average applied at two delays, $t = 0$ and $t = 1$ ps, with the corresponding spectra shown in Fig. 1e. To determine a specific anisotropy for a given feature, $\beta_2(\varepsilon)$ has been averaged over the spectral features as shown by the shaded regions in Fig. 1e. This yields values of $\beta_2 = -0.36 \pm 0.09$ and $\beta_2 = -0.11 \pm 0.12$ for the initial photoelectron peak at centred at $\varepsilon \sim 1.4$ eV ($\varepsilon_1$) and the lower energy peak centred at $\varepsilon \sim 0.8$ eV ($\varepsilon_2$), respectively.

**Assignment of photoelectron features.** There are two dominant pathways discussed for the initial $S_1$ state dynamics of PYP chromophores[19,20,22]. The ground state of $p$CK$^-$ is planar because of the $\pi$-conjugation over the $para$-substituent on the phenolate anion. Upon excitation to the $S_1$ state, an electron populates a molecular orbital with $\pi^*$ character, weakening the corresponding $\pi$-conjugation of the molecule and facilitating rotation about the bonds. Following $S_1$ photoexcitation, the molecule first rapidly relaxes to a local planar minimum with a geometry that is very similar to the Franck-Condon geometry (i.e. $S_0$ minimum). From this planar $S_1$ minimum, rotation about either the single bond, $\varphi_{SB}$, or the double bond, $\varphi_{DB}$, can occur as shown in Fig. 1a.

Figure 2 shows the relevant potential energy surfaces that have been calculated using high-level multireference methods along two different pathways connecting the $S_1$ planar minimum (PM) with the two minima on the $S_1$ surface. These two minima arise from rotation around $\varphi_{SB}$ or $\varphi_{DB}$ and their geometries are denoted as SB and DB, respectively, as shown inset in Fig. 2a. The calculated pathways connect the different minima via a linear interpolation in internal coordinates (LIIC) and as such account

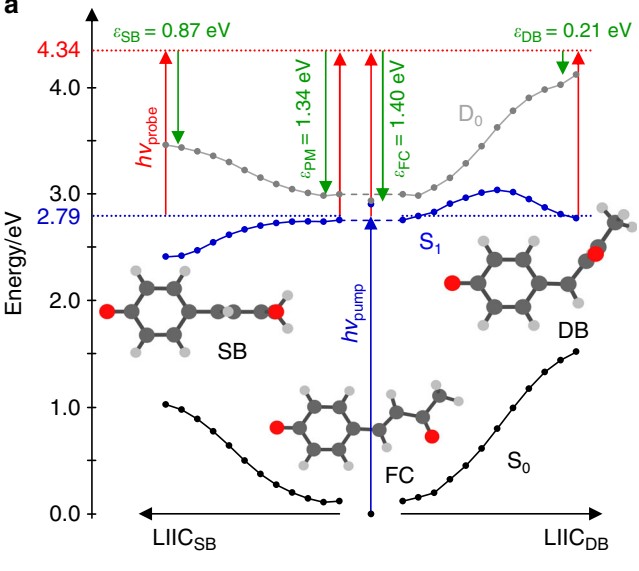

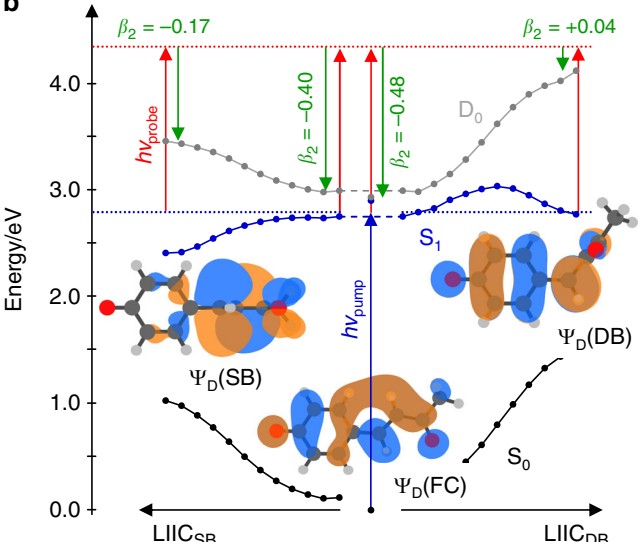

**Fig. 2 Calculated potential energy surfaces for isomerisation coordinates.** Calculated surface (XMCQDPT2) obtained from a linear interpolation in internal coordinates (LIIC) connecting: (left) the $S_1$ planar minimum (PM) and the $S_1$ minimum along the single-bond (SB) pathway; and (right) the $S_1$ PM and the $S_1$ minimum along the double-bond (DB) pathway. **a** Energetic analysis of excitation to the $S_1$ state. Structures indicate critical points on the $S_1$ potential energy surface, defined as the Franck-Condon structure (FC) and the $S_1$(SB) and $S_1$(DB) minima. Pump and probe photon energies are included and expected maximum kinetic energies for the resultant photoelectrons detached from the FC ($\varepsilon_{FC}$), PM ($\varepsilon_{PM}$), SB ($\varepsilon_{SB}$) and DB ($\varepsilon_{DB}$). **b** Analysis of the modelled photoelectron angular distributions corresponding to the same regions on the $S_1$ potential energy surface. The Dyson orbitals, $\Psi_D$, for SB and DB shown, $\Psi_D$(SB) and $\Psi_D$(DB), respectively. The Dyson orbitals for FC and PM are very similar and only $\Psi_D$(FC) is included. In the structures, carbon atoms are dark grey, hydrogen atoms are light grey, and oxygen atoms are red.

for the geometrical changes along the different points of the $S_1$ potential energy surface. While other nuclear displacements take place, the motion along either pathway is dominated by the rotations $\varphi_{SB}$ or $\varphi_{DB}$. Motion along the $\varphi_{DB}$ involves a barrier, while that along the $\varphi_{SB}$ coordinate is essentially barrierless. Our calculations are in reasonable agreement with previous theoretical

work, which suggested that $\varphi_{SB}$ rotation is more probable and also found a barrier along $\varphi_{DB}$ for related chromophores[20,22]. The main differences with previous theoretical works arise from the levels of theory used. Here, our main goal is to treat the $S_1$ excited state on the same footing as the $D_0$ final state to offer the most reliable energies to compare with the photoelectron spectra that are measured in the experiment. The vertical excitation energies of the $S_0$, $S_1$ and $D_0$ states are obtained from the same XMCQDPT2 calculation (see Computational Details for more information).

The photoelectron spectrum is determined by the difference in energy between the anionic ($S_1$) and neutral ($D_0$) states, as shown in Fig. 2a. Based on the calculated values, detachment of the Franck-Condon geometry (denoted FC in Fig. 2a) with a $h\nu =$ 1.55 eV probe will lead to electron signal extending to $\varepsilon_{FC} =$ 1.40 eV. The rapid initial relaxation to the $S_1$ planar minimum (PM) reduces this limit slightly to $\varepsilon_{PM} = 1.34$ eV. Rotation about $\varphi_{SB}$ and $\varphi_{DB}$ leading to the $S_1$ minima SB and DB is expected to lead to photoelectron signal extending to $\varepsilon_{SB} = 0.87$ eV and $\varepsilon_{DB} = 0.21$ eV. These limits are shown in Fig. 1e. It is important to note that the estimated uncertainty in the calculations is ~0.2 eV and that only the molecular geometry at each critical point was used to determine these energies. Additionally, the maxima of the photoelectron peaks are expected to be shifted to slightly lower energy compared to the predicted maximum kinetic energy because the potential energy at the minima is lower than at the initial excitation energy. For example, the predicted maximum signal for rotation about $\varphi_{SB}$ will occur in the range $0.43 < \varepsilon_{SB} <$ 0.87 eV. Hence, the calculated maximal values should be used as a guide only. Nevertheless, based on the potential energy surfaces in Fig. 2a, the agreement of the peak at $t = 0$ with the expected energy for the Franck-Condon (and $S_1$ planar minimum) geometry is excellent. At the later time of $t = 1$ ps, the broad peak centred at $\varepsilon \sim 0.8$ eV is consistent with a twisted intermediate that has undergone rotation about $\varphi_{SB}$. This peak is not consistent with rotation about $\varphi_{DB}$ as the spectral maximum of DB is expected at $\varepsilon < 0.21$ eV. Hence, based solely on energetic arguments, the dynamics involving the peaks at $\varepsilon_1 \sim$ 1.4 eV and $\varepsilon_2 \sim 0.8$ eV correspond to dynamics involving rotation about the single bond.

Rotation about specific bonds also leads to differing electronic structures: adiabatically, a change in nuclear configuration is associated with an instantaneous adaptation of the underlying electronic structure. That is to say, the character of the valence orbitals at a given molecular geometry should be reflected in the laboratory-frame PADs and these may be expected to be different for the two different isomerisation pathways. Such changes can be quantitatively analysed by computing the Dyson orbital, $\Psi_D$, for the key structures along the reaction coordinate. The Dyson orbital can be thought of as the one-electron wavefunction describing the electron that is being photodetached. Krylov and coworkers have shown that PADs can be conveniently calculated from $\Psi_D$ yielding computed $\beta_2(\varepsilon)$ trends[27,28]. We have previously shown that computed $\beta_2(\varepsilon)$ are in satisfactory agreement with experimental ones for several molecular anions in their ground state, including para-substituted phenolate anions, which $p$CK$^-$ is a derivative of[29,30]. Moreover, we showed that PADs are also sensitive to subtly differing electronic structure when a short alkyl chain (ethyl) lies either in the plane of the phenolate ring or perpendicular to it[30]. We have now extended these calculations to predict the $\beta_2(\varepsilon)$ for detachment from the $S_1$ excited state of $p$CK$^-$.

Figure 2b shows $\Psi_D$ for key critical geometries: the Franck-Condon geometry, $\Psi_D$(FC), and the two $S_1$ minima associated with a rotation about $\varphi_{SB}$ and $\varphi_{DB}$: $\Psi_D$(SB) and $\Psi_D$(DB). Laboratory-frame PADs were calculated based on these $\Psi_D$, with the neutral $D_0$ ground state as the final state. The computed $\beta_2$

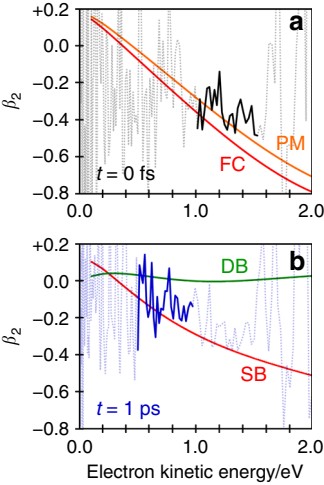

**Fig. 3 Photoelectron anisotropy parameters along adiabatic coordinate.** $\beta_2$ parameters at $t = 0$ (**a**) and $t = 1$ ps (**b**) with experimental (black and blue, respectively) and computed (red, orange, and green) values shown. For clarity, the region over which the experimental and computed $\beta_2$ values are comparable is highlighted in the experimental data. Dyson orbitals corresponding to the critical points on the potential energy surface are shown in Fig. 2b.

values can be directly compared to the measured values (Fig. 1f). The simplest comparison can be done by averaging the computed $\beta_2$ values over the same energy range as for the experimental results. This yielded computed anisotropy parameters for key geometries of $\beta_2 = -0.48$ (FC), $\beta_2 = -0.40$ (PM), $\beta_2 = -0.19$ (SB) and $\beta_2 = +0.04$ (DB). These can be directly compared with experimental values of $\beta_2 = -0.36$ and $\beta_2 = -0.11$ for the initial peak at $t = 0$ and the peak at $t = 1$ ps. Such a comparison suggests that the signal in $\varepsilon_1$ arises from FC and PM, while the signal in $\varepsilon_2$ arises from SB. A more useful comparison is based on the trends of $\beta_2(\varepsilon)$. From Fig. 3a, the measured $\beta_2(\varepsilon)$ for the peak at $t = 0$ is in reasonable quantitative agreement with the $\beta_2(\varepsilon)$ computed from $\Psi_D$(FC) and $\Psi_D$(PM). From Fig. 3b, $\beta_2(\varepsilon)$ for the photoelectron peak at $t = 1$ ps is in reasonable quantitative agreement with $\beta_2(\varepsilon)$ computed from $\Psi_D$(SB). In contrast, the agreement for this feature with $\beta_2(\varepsilon)$ computed from $\Psi_D$(DB) is poor and qualitatively has the wrong sign and trend. Other points along the $\varphi_{DB}$ coordinate, including at the barrier, yielded predominantly positive $\beta_2(\varepsilon)$ values, similar to that predicted from $\Psi_D$(DB), and thus also qualitatively different to the observed experimental trends. We conclude that the signal in the $\varepsilon_2$ spectral range is a direct measure of the single bond pathway rather than the double bond one upon photoexcitation to $S_1$ of $p$CK$^-$ and that the dynamical changes between $\varepsilon_1$ and $\varepsilon_2$ reflect adiabatic motion between the FC/PM region to the SB minimum on the $S_1$ excited state. This conclusion is consistent with the energetic arguments made earlier.

Overall, the agreement between predicted and measured $\beta_2(\varepsilon)$ is almost quantitative, especially given that these are based on single geometries that do not account for other nuclear motions (either thermal or photoinduced), which will tend to make the PADs more isotropic. Moreover, the calculation of the PADs employs some key approximations. In particular, the outgoing wave is treated as a plane wave and thus assumes no interaction of the photoelectron with the neutral core. In the present case, this may be a poor approximation because the neutral $p$CK core has a large permanent dipole moment. Despite these limitations, the agreement is very good, especially in terms of the trends of $\beta_2$ with $\varepsilon$ as see in Fig. 3.

Inspection of $\Psi_D$ in Fig. 2b provides intuitive chemical insight about how the PADs reflect the changes in electronic structure along the isomerisation coordinate. Specifically, we have previously used a simple Hückel model to interpret changes in the excited state energies and character for a series of *para*-substituted phenolate anions[29]. As *p*CK$^-$ belongs to this family, similar arguments apply here. The $S_1$ and $S_2$ states can be considered as linear combinations of molecular orbitals localised on the phenolate ring and the π-conjugated *para*-substituent. From Fig. 2b, rotation about $\varphi_{SB}$ leads to a localisation of $\Psi_D(SB)$ onto the π-conjugated substituent. Locally, $\Psi_D(SB)$ is therefore associated with a planar π-conjugated system and this is expected to lead to $\beta_2 < 0$, similar to that predicted for $\Psi_D(FC)$[26]. In contrast, following the rotation about $\varphi_{DB}$, $\Psi_D(DB)$ becomes delocalised over a non-planar moiety. Such a molecular orbital is expected to yield $\beta_2 \sim 0$, as previously seen in the ground electronic state of *para*-ethylphenolate[30]. Hence, despite the complex nature of lab-frame PADs, simple arguments provide an intuitive view of the electronic structure changes associated with the isomerisation coordinate. Hence, without the need to perform high-level calculations, the observed PADs can provide qualitative insight into the changes in valence-bonding along the isomerisation coordinate.

## Discussion

Based on the spectral and angular distributions, the photoelectron signal at $\varepsilon_1$ is assigned to the signature of FC and PM and that at $\varepsilon_2$ to the twisted intermediate following rotation about the single bond, SB. The dynamics associated with this evolution is shown in Fig. 1d. The coherence observed shows a nuclear wavepacket moving on the excited state surface from the $S_1$ planar minimum past the SB minimum and back again with a period of ~400 fs. Note that the vibrational modes that comprise this wavepacket are not necessarily the Franck-Condon active modes. The dominant FC modes are likely to stretch the C–C bonds as the excitation involves a π* ← π transition. These are high frequency modes that lead to very rapid dynamics from the FC towards the $S_1$ minimum. This motion then evolves into the modes that lead to isomerisation. The observed oscillation is in agreement with excited state molecular dynamics simulations of *p*CK$^-$ that have predicted a similar oscillation[20]. Only a single oscillation is observed, presumably as a result of the dephasing to other modes (i.e. internal vibrational energy redistribution).

The time-resolved photoelectron spectroscopy experiment by Zewail and coworkers similarly noted energetic shifts and associated dynamics, following photoexcitation at $h\nu = 3.10$ eV[16]. Excitation at 3.10 eV is above the adiabatic detachment energy and probably also above the barrier to double bond rotation. In their experiments, autodetachment from the $S_1$ state (characterised by electrons at low $\varepsilon$) was a prominent feature, which could swamp any signatures of the dynamics associated with double bond rotation that might have been occurring. We also observe a very small fraction of autodetachment (≲4%), enabled by the finite temperature (~300 K) and the spectral width of the pump pulse. Additionally, Zewail and coworkers observed an oscillation in the high kinetic energy window, similar to that observed in $\varepsilon_1$, but not the out-of-phase oscillation at lower energy ($\varepsilon_2$), probably because of contamination by autodetachment[16]. Dynamics involving isomerisation were also observed in a recent study on a closely related PYP chromophore anion in which the ketone is replaced by an ester group[31]. These dynamics were in competition with internal conversion to a non-valence state of the anion. Such dynamics are not observed here highlighting that even small chemical changes can have a marked impact on the excited state dynamics.

Finally, Fig. 1b and e shows a peak at very low $\varepsilon$ ($\varepsilon_3$) in the time-resolved photoelectron spectra. This spectral peak could arise from double bond rotation. The maximum expected energy for photodetachment from DB, $\varepsilon_{DB} = 0.21$ eV, which would be consistent with this peak. It is not informative to analyse the PADs for this channel because they are at too low kinetic energy, where the PADs are generally expected to be isotropic. However, a number of observations may suggest a different origin of the $\varepsilon_3$ signal. Firstly, the formation of the DB minimum involves motion along the $\varphi_{DB}$ coordinate (Fig. 2a) and should lead to photoelectron signals that evolve continuously from FC/PM to the DB minimum; but this is not observed. Secondly, the oscillation frequency of the integrated signal in $\varepsilon_2$ and $\varepsilon_3$ is essentially identical (Fig. 1d); one might expect that the period of motion to differ slightly between the two coordinates. Thirdly, if this signal was attributed to DB rotation, then the minimum of the photoelectron signal in $\varepsilon_3$ would arise because the probe photon energy was insufficient to access the final neutral state ($D_0$)[32]. In that case, the oscillation should be observable in the total photoelectron signal, but no such changes are seen (Supplementary Fig. 4). Instead, we suspect that the signal in $\varepsilon_3$ comes about because the probe can access the first excited state of the neutral, $D_1$. This excited state can be seen in the photoelectron spectrum at higher photon energies (see Supplementary Fig. 2). According to our calculations, the vertical energy difference between the SB intermediate on $S_1$ and the $D_1$ is 1.3 eV, suggesting that it could be accessed with the 1.55 eV probe. Nevertheless, the assignment of this feature remains somewhat uncertain and we cannot exclude that concomitant dynamics about $\varphi_{DB}$ are taking place on the $S_1$ state over the first picosecond. It would be useful to probe the dynamics with a higher energy photon. However, this comes with added complications of possible excitations from the $S_1$ to higher-lying excited states of the anion.

In summary, we have probed the geometric and electronic structure of a polyatomic molecule using time-resolved photoelectron imaging. In combination with calculations beyond the Franck-Condon region, we can identify specific signals that arise from an isomerisation coordinate involving rotation about the single bond in *p*CK$^-$. The photoelectron signal provides information about changes in the energies of potential energy surfaces along an intramolecular coordinate, while the photoelectron angular distributions capture the changes in electronic structure that arise from such an isomerisation. While we can conclusively identify single-bond rotation, we cannot exclude that double-bond rotation may be occurring also, because its photoelectron signatures are not captured well in the current experiments. To the best of our knowledge, this presents the first study in which lab-frame photoelectron angular distributions have been tracked along a non-dissociative adiabatic coordinate and that have been quantitatively modelled. These methods provide a basis for probing adiabatic dynamics in large molecular systems.

## Methods

**Experimental details.** Experiments were performed on an anion photoelectron imaging spectrometer[33]. Anions were produced by negative-mode electrospray ionization of *p*CK in methanol at pH ~ 10 and transferred into vacuum where they were stored in a ring-electrode trap, thermalized to ~300 K, and unloaded into a time-of-flight mass spectrometer at 100 Hz. Mass-selected anions were intersected by a pair of delayed femtosecond pulses at the centre of a velocity-map imaging spectrometer, which monitored the velocity vectors of the emitted photoelectrons. Probe pulses used the fundamental of a Ti:Sapph (450 μJ pulse$^{-1}$) and pump pulses were generated by 4th harmonic generation of idler of an OPA (5 μJ pulse$^{-1}$) and interacted with the sample unfocussed (beam diameter ~ 3 mm). Pump and probe polarizations were set parallel to the detector. The temporal instrument response is 100 fs and times are accurate to better than ±10 fs. Raw photoelectron images were analysed using polar onion peeling[34], which recovers the 3D electron velocity distribution from the 2D projection measured on the position sensitive detector

(see Supplementary Methods and Supplementary Fig. 5). This analysis yields photoelectron spectra and PADs that were calibrated using the photoelectron spectrum of iodide.

**Computational details**. The energetic minima corresponding to $FC(S_0)$, the planar $S_1$ state and $SB(S_1)$ and $DB(S_1)$ were first located at the SA2-CASSCF(12,11)/6-31G* level of theory (see Supplementary Fig. 6 and Supplementary Table 1)[35]. Linear interpolation in internal coordinates (LIIC) pathways were obtained to link the different critical points. An LIIC pathway gives the most straightforward path from a given molecular geometry to a different geometry by interpolating a series of geometries in between, using internal (not Cartesian) coordinates (see for example ref. [36]). It is important to note that *no* reoptimisation of the molecular geometries is performed along these pathways, implying that LIIC pathways do not correspond to minimum energy paths, *per se*. In particular, the barriers observed along LIIC pathways are possibly higher than the actual barriers one would obtain by searching for proper transition states. LIICs, however, offer a clear picture of the possible pathways between critical points of potential energy surfaces and allow to predict photophysical and photochemical processes that a molecule can undergo. The electronic energy of the $S_1$, $S_2$ and $D_0$ states were recalculated at all points along the LIICs using multi-state extended multi-configurational quasi-degenerate perturbation theory (MS-XMCQDPT2)[37] to correct for the lack of dynamic correlation at the SA-CASSCF level. The (aug-)cc-pVTZ basis set was used where the augmented function was only affixed to the oxygen atoms[38]. The $D_0$ was calculated through addition of an orbital characterized by an extremely diffuse p-function ($\alpha = 1E{-}10$) in the active space and included in the 6 state averaging procedure to mimic detachment to the continuum[39–41]. A rigid shift was applied to match the $S_0$–$D_0$ energy to the experimentally determined vertical detachment energy of 2.94 $\pm$ 0.05 eV at the Franck-Condon geometry. A DFT/PBE0-based one-electron Fock-type matrix was used to obtain energies of MCSCF semi-canonical orbitals used in perturbation theory as done elsewhere[39–41].

The Dyson orbitals for critical geometries were calculated using EOM-EE/IP-CCSD/6-31+G** [27,28,42,43] and the PADs were modelled using ezDyson v4[44]. EOM-EE-CCSD calculations with the 6-31+G** basis set were also used to determine the vertical excitation energies of the first excited state of the neutral, $D_1$, at the minimum energy geometries on the $S_1$ surface.

The initial SA-CASSCF calculations were performed with Molpro 2012[45], XMCQDPT2 calculations were carried out using the Firefly quantum chemistry package[46] and EOM-EE/IP-CCSD calculations used QChem 5[47].

## Data availability
All data is available at https://doi.org/10.5281/zenodo.3750967.

## Code availability
Polar onion peeling is available at https://www.github.com/adinatan/PolarOnionPeeling (Matlab version) or http://www.verlet.net/research.html (Labview version).

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

## Acknowledgements

We thank James Bull and Anastasia Bochenkova for allowing access to computing facilities and Anna Krylov for helpful discussions regarding Dyson orbital calculations. This work has been funded by the European Research Council under Starting Grants 306536 and 803718.

## Author contributions

J.R.R.V. conceived the project. C.S.A. performed most of the experiments and calculations with help from J.R.R.V. and B.F.E.C., respectively. C.S.A. analysed the data and all discussed the results. J.R.R.V. wrote the paper with input from C.S.A. and B.F.E.C.

## Competing interests

The authors declare no competing interests.
