## [Peer Review File · Nature Communications]

Reviewers' comments:

Reviewer #1 (Remarks to the Author):

The authors have performed time-resolved photo-electron spectroscopy and time-resolved photo-electron imaging on a derivative of the Photoactive Yellow Protein chromophore in the gas-phase. After pumping the deprotonated chromophore to the π - π^* S1 state with a 444 nm laser pulse, the kinetic energy and velocity vectors of electrons ejected from the excited chromophore with a 800 nm probe pulse, were measured at various pump-probe delays. The results of these measurements reveal that two (three, if also the signal at ~ 0.1 eV is included) regimes of photo-electrons are observed with different lifetimes, energies and emission-directions with respect to the polarization of the pump pulse (beta2).

To interpret these results, the authors have computed interpolated S1 potential energy profiles for rotation around two torsion angles: the single bond (SB) adjacent to the phenolate, and the (formal) double bond (DB). Based on the energy gaps between the S1 state and the photo-ionized D0 state at the minima of these profiles, the authors attribute the higher energy photo-electron signal at ~ 1.4 eV to photo-ionization from the planar S1 minimum near the Franck-Condon region. The second signal at ~ 0.8 eV is attributed to photo-ionization from the SB-twisted S1 minimum. Although the energy and beta2 parameters of the third (weak) signal (epsilon3) would match (in my opinion) with electrons photo-ionized from the DB minimum, the authors dismiss the latter because no evolution from the S1 minimum to this DB minimum is observed in the time-resolved photo-electron spectra (figure 1). As I try to explain below, I may have not understand this argument. Nevertheless, based on the good agreement between measured and computed photoelectron energies and vectors, the authors conclude that, when excited without too much excess energy (which was the case in Lee et al. Proc. Natl.Acad. Sci. USA, 2006, 103, 258), the isolated chromophore relaxes exclusively into the single bond twisted minimum, in line with the predictions from previous computations.

While I consider in particular the experiments of impressive quality, and the scientific question about what bond isomerizes in the PYP chromophore timely and relevant, I am not yet totally convinced that the experiments rule out rotation around the DB as an additional excited state relaxation channel. I explain below why.

First, I probably have not understand why the photo-electron signal around 0.1 eV cannot be attributed to the DB-twisted S1 minimum. I agree that the 2D spectra show no evident evolution connecting the higher energy signal of the S1 planar minimum to the signal at 0.1 eV, but I would also not expect to see that, because of the barrier between the S1 planar and DB-twisted minima. The barrier crossing might be stochastic (kinetic) and the population at the barrier could thus be very

small, despite transitions. Also, the signal at epsilon 3 still seems to rise in concert with epsilon 2 at the expense of epsilon 1 in figure 1d, which also might hint at transitions in two new minima, rather than one.

Furthermore, I also do not understand why the signals of epsilon 2 and epsilon 3 seem to exceed the maximum signal of epsilon 1 in figure 1d. At least it seems to me the sum of the signals is not constant. Does this imply that the photo-ionization cross-sections depend strongly on the chromophore geometry? Do the calculations provide insight?

My second concern about the interpretation of the photo-electron spectra is the (potentially) limited accuracy of the computational results, in particular for photo-ionization from the DB-twisted minimum. While the xMCQDPT2 method is arguably today's state-of-the-art for computing excitation energies, what worries me are the (small) discrepancies with previous results. For example, Groenhof and co-workers computed the interpolated energy profiles between the planar S1 minimum and the SB and DB twisted minima at the xMCQDPT2/SA2-CASSCF(12,11)/cc-pVTZ level of theory and find that the DB twisted minimum is about 0.2 eV below the S1 planar minimum (Knoch et al. *Comp. Theory Chem.* 2014, 1040-1041, 120-125). In contrast, the profiles computed by the authors suggest that the DB S1 minimum has the same energy as the planar S1 minimum. Also the energy gaps at the minima seem rather different compared to those previous xMCQDPT2 calculations: In figure 2 of the present work, the gap at the DB-minimum, it is roughly 1.1 eV, whereas Knoch et al. report 0.6 eV, a substantial difference. While it is not clear a priori which results are more accurate, or why, the discrepancy suggests nevertheless that if, the DB minimum were indeed 0.2 eV or more below the S1 planar minimum, the S1->D0 energy gap could potentially exceed the 1.55 eV of the probe pulse and hence be undetectable in the measurements. Thus, before the DB bond rotation can be ruled out based on the computations, I feel it is important to have an idea on whether the expected accuracy of the computed results is sufficiently high for such conclusion.

I also had some difficulties understanding the comparison between the calculated and measured photo-electron energies. From the graph in Figure 2, I roughly estimate that the SB minimum is around 2.4 eV at the xMCQDPT2 level of theory, while the D0 state is (again rough estimate) at 3.5 eV. Thus, unless I made a mistake in reading these numbers, the energy gap is 0.9 eV. With a probe energy of 1.55 eV, the ejected electron would have a kinetic energy of 0.65 eV. However, in the text the authors write (line 154) that the energy is 0.87 eV. While the latter would indeed be very close to the experimental result of ~0.8 eV, the 0.65 estimate is a little farther off, possibly even beyond the expected error of the underlying theoretical method? Unless I made a mistake, I suggest the authors to check their numbers and, if needed, adjust the text and/or figure.

Also, please include the structures, and energies as SI, as well as the CASSCF profiles. This would facilitate comparison to previous (and future) works.

The authors observe interesting oscillations in the photo-electron signals and attribute these oscillatory features to the wavepacket moving between the SB-twisted and the planar S1 minima. While this may be the case in vacuum, my understanding of oscillatory features in transient spectroscopy signals is that ultra-short pulses excite vibrationally coherent populations in the S1 state (and also S0, but that was not probed here, see e.g. Bardeen et al. Phys. Rev. Lett. 1995, 75, 3410). Thus the coherent vibrations would need to be Franck-Condon-active. Is the SB rotation coordinate also Franck-Condon active? Does the period of the (single) oscillation of the 1.4 eV signal match the vibronic progression?

The authors link their observations to the oscillations seen in the fluorescence upconversion experiments of Mataga and co-workers on the wtPYP protein (Chosrowjan et al. J. Phys. Chem. B 2004, 108, 2686) . However, experimental and also computational evidence suggests that in the protein, only the DB-rotation pathway is active. If that is true, I do not see how the initial photo-dynamics in PYP can be the same as in the isolated chromophore or that the protein would have no influence (line 213-216).

The authors mention that the signal at 0.8 eV decays with a timeconstant of 120 ps. Thus, I suppose the signal was measured for much longer time delays (at least up to 100s of ps) than displayed in figure 1. I suggest to include the complete data set as SI.

What is the origin of the slow 120 ps decay process?

Summarizing this is a very interesting and timely study that deserves to appear in Nature Communications as it would interest a broad community of chemists. However, the interpretation of the photoelectron spectra seems to depend strongly on the underlying computational model, which might have some accuracy issues. I am of the opinion therefore that the latter should be addressed before the manuscript can be published.

Reviewer #2 (Remarks to the Author):

General: this study reports interesting results from PAD measurements. This paper is written in a certain style using specific lab jargon. Nature Communication tends to the more general reader who might find these results not interesting, unless results and methodology are explained to an educated non-expert audience. The authors are strongly encouraged to do so, and do not leave it to

the reader to consult the literature to put together the validity of the claims made here. Otherwise, this manuscript could be easily submitted to a more specialized journal.

It starts already with the photoelectron spectra (Figure S1). Would it not be possible for the authors to define VDE and ADE in the legend? Would it not further be possible to spell out eKE?

The main text mentions that the supplemental material provides details on photoelectron images ..., it does not. It does not even show a single detector image, not to mention a pipeline for data processing.

The supplemental material reports that PAS were recorded with ns laser pulses. It is not clear why a photon energy of 3.45 eV avoids excitation of the excited state (to wit: "excitation is the least resonant with excited states of the anion"). Maybe the authors mean "photoproduct excitation".

Further, along this line, the main text mentions 2.79 eV (444 nm) excitation with femtosecond pulses rather than using the 3.45 eV ns laser pulses as mentioned in the supplemental material. Would it not be possible to add one sentence to make this (trivial) distinction between the ns and fs laser pulses clear for the reader to avoid confusion? For pCA in PYP it is known that the ground and excited state cross sections are particularly favorable around 450 nm. This should be clearly stated and explained.

Define Franck Condon (S0), SB(S1), DB(S1) and D0. A few words on LIIC pathway determination would be adequate.

Explain in plain words why the (aug)-cc-pVTZ basis affixed to the oxygens corrects for dynamic correlation (what dynamic correlation is relevant here), and why is this necessary.

Explain in a few sentences "onion peeling". More generally: The reader should have a chance to understand how lab and molecular frames can be connected without digging into the literature.

Discussion of Fig. 1 and Fig. 2 is overly complicated and full of lab jargon. For example, there is this sentence: Figure 2b shows Ψ_D for key critical geometries, YET Ψ_D is not mentioned anywhere in Figure 2b.

Further there is the claim that lab frame PADs were calculated from the Ψ_D : Why not to show the calculated, time-resolved lab frame PADs and compare them with the observed PADs? Maybe they cannot be compared. If so, mention this.

The main conclusion of this paper is presented in a relative sentence following a main sentence that has nothing to do with single bond rotation: "...expected energy for the Franck Condon ..., while the broad peak ...". The authors are encouraged to learn how to present and highlight their main scientific findings to the non-expert. This publication should not be a lab manual.

Fig. 2 is intriguing, but again, the discussion is strangely disconnected from the data. It is quite interesting to note that the mechanism in Fig. 2 a might indeed coincide with ultrafast events

observed in time-resolved crystallographic measurements on PYP (Pande et al., Science 2016). Maybe the authors comment on this.

As non-Born-Oppenheimer dynamics is mentioned in the introduction, it requires some explanation why it appears here.

For a successful publication in Nature communications, this reviewer recommends:

- avoid lab jargon
- argue for the laymen
- spell out the reasons why things were done, don't leave this to the imagination of the reader
- explain data processing and data analysis (in the supplemental material)
- in addition to pointing to literature, briefly explain the other methods employed in this paper.
- highlight and present main scientific findings separately, in main sentences, and in plain words. Avoid trying to explain two things in the same sentence.
- explain the data with findings from theoretical calculations by pointing out specific features. The more the reader understands the first time, the more valuable the results become.

Reviewer #3 (Remarks to the Author):

Review of manuscript by Prof Verlet and co-workers entitled "Geometric and electronic structure probed along the isomerisation coordinate of a photoactive yellow protein chromophore"

The present submission is one in a series by the Verlet group related to slightly different model chromophores of the Photoactive Yellow Protein (PYP). They include (as far as I have seen) in addition to the present submission:

[1] "Ultrafast valence to non-valence excited state dynamics in a common anionic chromophore". Nature Com. (2019)

[2] "Fingerprinting the Excited State Dynamich in Methyl Ester..." J. Phys. Chem A. Just accepted manuscript. (2020)

Although they address slightly different aspects of the photo physics of the PYP chromophore, the two former papers already contain parts (although not all) of information which is addressed in the present submission.

Why no reference to [1] is given in the present paper is unclear to me.

General comments:

The paper is well-written and for the most part clear to follow.

The introduction of the present submission refers to the utilization of the time-resolved photoelectron imaging technique. In my mind, the application of this technique alone does not warrant a Nature Comm. paper. The combined electronic and nuclear dynamics is here indeed beautifully demonstrated by the time-resolved photoelectron imaging technique, but the technique is not brand new, and several reviews accounting on its various aspects exist (Stolow, Zanov, Toshinori, Neumark).

In the second paragraph, it is argued that the PYP protein is interesting and in particular the question about which specific bond isomerization occurs. It is true that the chromophore is particularly small and it is used as a model system for experiments along with theoretical studies. To the general readership of this Journal, this PYP model chromophore, however is perhaps not all that interesting. Other bio-chromophores for vision and energy harvesting might be more stimulating to read about.

So, whether the system and the applied methodology is sufficiently new, groundbreaking, and interesting is to me questionable. That is not to say that the quality of the work is not high, I think it is!

Specific comments:

In figure 2a I need a discussion on why $\epsilon_{\text{SB}} = 0.87$ eV is used. If within, the time available between the pump and probe pulse, intramolecular vibrational redistribution (IVR) takes place, energy will be transferred to vibrations, which then is not available for the outgoing electron. Thus, the red arrow (upper left in figure 2a) could start on the S1 curve as well and hence give an ϵ_{SB} which is about

0.4 eV lower than 0.87 eV. The same goes for figure 2b. I think this needs a comment in the paper at least. To see if IVR has fully occurred, one could look at the kinetic-energy spectrum after for example 100 ps.

The paragraph (line 192-196) discussing figure 2b did not provide me with intuitive insight. Could the authors perhaps elaborate on this?

Reviewers' comments:

Reviewer #1 (Remarks to the Author):

The authors have performed time-resolved photo-electron spectroscopy and time-resolved photo-electron imaging on a derivative of the Photoactive Yellow Protein chromophore in the gas-phase. After pumping the deprotonated chromophore to the $\pi\text{-}\pi^*$ S1 state with a 444 nm laser pulse, the kinetic energy and velocity vectors of electrons ejected from the excited chromophore with a 800 nm probe pulse, were measured at various pump-probe delays. The results of these measurements reveal that two (three, if also the signal at ~ 0.1 eV is included) regimes of photo-electrons are observed with different lifetimes, energies and emission-directions with respect to the polarization of the pump pulse (beta2).

To interpret these results, the authors have computed interpolated S1 potential energy profiles for rotation around two torsion angles: the single bond (SB) adjacent to the phenolate, and the (formal) double bond (DB). Based on the energy gaps between the S1 state and the photo-ionized D0 state at the minima of these profiles, the authors attribute the higher energy photo-electron signal at ~ 1.4 eV to photo-ionization from the planar S1 minimum near the Franck-Condon region. The second signal at ~ 0.8 eV is attributed to photo-ionization from the SB-twisted S1 minimum. Although the energy and beta2 parameters of the third (weak) signal (epsilon3) would match (in my opinion) with electrons photo-ionized from the DB minimum, the authors dismiss the latter because no evolution from the S1 minimum to this DB minimum is observed in the time-resolved photo-electron spectra (figure 1). As I try to explain below, I may have not understand this argument. Nevertheless, based on the good agreement between measured and computed photoelectron energies and vectors, the authors conclude that, when excited without too much excess energy (which was the case in Lee et al. Proc. Natl.Acad. Sci. USA, 2006, 103, 258), the isolated chromophore relaxes exclusively into the single bond twisted minimum, in line with the predictions from previous computations.

While I consider in particular the experiments of impressive quality, and the scientific question about what bond isomerizes in the PYP chromophore timely and relevant, I am not yet totally convinced that the experiments rule out rotation around the DB as an additional excited state relaxation channel. I explain below why.

At this point, we would like to comment that we agree with the referee. It was never our intention to categorically dismiss the double-bond rotation, and we agree with the referee that the manner in which we had written the paper may have left the reader with that impression although it was not our intent. The main claim that we do insist on is that we can measure single-bond rotation through the photoelectron signals that we have analysed. That is not to say that we are 100% sure that double-bond rotation is not occurring, but these signals may appear in other spectral ranges. With this in mind, we will address the comments below.

First, I probably have not understand why the photo-electron signal around 0.1 eV cannot be attributed to the DB-twisted S1 minimum. I agree that the 2D spectra show no evident evolution connecting the higher energy signal of the S1 planar minimum to the signal at 0.1 eV, but I would also not expect to see that, because of the barrier between the S1 planar and DB-twisted minima. The barrier crossing might be stochastic (kinetic) and the population at the barrier could thus be very small, despite transitions. Also, the signal at epsilon 3 still seems to rise in concert with epsilon 2 at

the expense of epsilon 1 in figure 1d, which also might hint at transitions in two new minima, rather than one.

The energies match up – that we agree with. In terms of angular distributions, for low energy electrons, the b_2 parameter is essentially zero, regardless of which orbital the electron leaves from. So this measure is not particularly sensitive. We have arrived at our interpretation that the low energy signal is likely not due to the DB rotation because we feel that the dynamics over the transition state on the S_1 state should be observable. There is a priori no reason why this should not appear in the photoelectron spectra, even if it was a short-lived transient. We (and others) have seen spectral signatures of transient populations with lifetimes <20 fs before. The second reason is that the dynamics of the low energy peak mirrors the dynamics of the SB rotation. It would seem rather remarkable that the wavepacket along either bond rotation would be identical. Finally, we know that the D_1 state of the neutral is expected to be accessible by the probe photon. Of course, none of these provide absolute proof and so we agree that we should soften the tone and make this a discussion point in which we also consider DB rotation rather than exclude it as a possibility. Unfortunately, probing with higher photon energies will not provide more insight in the present case because the probe would be resonant with the $S_2 \leftarrow S_1$ transition, which would make the overall interpretation far more complex.

Furthermore, I also do not understand why the signals of epsilon 2 and epsilon 3 seem to exceed the maximum signal of epsilon 1 in figure 1d. At least it seems to me the sum of the signals is not constant. Does this imply that the photo-ionization cross-sections depend strongly on the chromophore geometry? Do the calculations provide insight?

This is not the case and we apologise that it appeared this way. The data originally presented in Figure 1d had a minor error in it which we now have rectified (the data in Figure 1b is correct). The total integrated photoelectron signal is almost constant over the first picosecond and importantly does not show the oscillation. There is a very small decay of the signal but this is small. The signals in epsilon 2 and 3 do not exceed that in epsilon 1. It is also important to recognise that the two spectra (epsilon 1 and 2) spectrally overlap a little and this inevitably causes problems in any analysis. We specifically refrained from over-analysing Figure 1d for these reasons and it is primarily included to show that the vibrational wavepacket is visible and that epsilon 1 is out of phase with 2 and 3 (which are in phase with each other). We have added a discussion to make this clear.

My second concern about the interpretation of the photo-electron spectra is the (potentially) limited accuracy of the computational results, in particular for photo-ionization from the DB-twisted minimum. While the xMCQDPT2 method is arguably today's state-of-the-art for computing excitation energies, what worries me are the (small) discrepancies with previous results. For example, Groenhof and co-workers computed the interpolated energy profiles between the planar S_1 minimum and the SB and DB twisted minima at the xMCQDPT2/SA2-CASSCF(12,11)/cc-pVTZ level of theory and find that the DB twisted minimum is about 0.2 eV below the S_1 planar minimum (Knoch et al. Comp. Theory Chem. 2014, 1040-1041, 120-125). In contrast, the profiles computed by the authors suggest that the DB S_1 minimum has the same energy as the planar S_1 minimum. Also the energy gaps at the minima seem rather different compared to those previous xMCQDPT2 calculations: In figure 2 of the present work, the gap at the DB-minimum, it is roughly 1.1 eV, whereas Knoch et al. report 0.6 eV, a substantial difference. While it is not clear a priori which results are more accurate, or why, the discrepancy suggests nevertheless

that if, the DB minimum were indeed 0.2 eV or more below the S₁ planar minimum, the S₁->D₀ energy gap could potentially exceed the 1.55 eV of the probe pulse and hence be undetectable in the measurements. Thus, before the DB bond rotation can be ruled out based on the computations, I feel it is important to have an idea on whether the expected accuracy of the computed results is sufficiently high for such conclusion.

There are differences in the methods used by others and the ones here. The main reason for this is that we aimed to calculate not only the S₁ state surface, but also the D₀ final state in a *single* calculation. This offers the most reliable energy differences between these states that ultimately determines the photoelectron spectrum. It is therefore not unexpected that small differences are present, but we feel that, overall, our agreement with previous work is very good. To go into a little more detail:

1) Our LIIC at the SA-CASSCF level of theory (that has been used to compute the XMCQDPT2 pathways) is in excellent agreement with the pathway computed by Groenhof and coworkers in the reference cited (10.1016/j.comptc.2014.04.005). We compared the different critical points and all our S₁ and S₀ energies are in close agreement (total electronic energy within less than 50 meV). In other words, our energies, at the SA-CASSCF level of theory, are in agreement. Groenhof and coworkers also found that the bigger energy difference between the minima is between S₁ min and DB min.

2) The energy differences observed between the different minima at the SA-CASSCF level of theory are washed out when switching on the PT2 level of theory. In the Groenhof and coworkers paper, the S₁ min (planar) is only 0.1 eV above the DB min, and 0.05 eV above the SB min. Considering that all these methods have an accuracy of ~0.2 eV (at best), they have to be considered as almost degenerate – in agreement with our XMCQDPT2 calculations for the S₁ min and the DB min. It is true that our SB min is a little lower in energy, but considering the accuracy of computational methods, the difference is not abnormal.

3) It is important to recognise the differences between the strategy used in the work by Groenhof and coworkers and the present work (XMCQDPT2): We use (1) a different active space and state-averaging and (2) we have a different (larger) basis set ((aug)-cc-pVTZ vs cc-pVDZ). Both can contribute to the difference highlighted above. On top of this, we have the presence of the orbital for the neutral state, which can be described within the *same* calculation as the one for the anion states (i.e., we do not perform two sets of calculations). Given these subtle differences, it is hard to say which calculations are of higher quality. Our larger state averaging combined with the description of the neutral state may lower the overall accuracy of the energies, but critically for the current paper, the energy gaps to the neutral are expected to be trustworthy. Overall, we might expect that the D₀ state would be shifted by the same amount as the S₁ state in our calculation, i.e., the 0.2 eV difference should be added to both S₁ and D₀.

Ultimately, we believe that we have used the most appropriate computational methodology for the current application (i.e. comparison to photoelectron spectra) but certainly appreciate that small differences are present. We have included a short statement to highlight this and have included more information in the supporting information (see below).

I also had some difficulties understanding the comparison between the calculated and measured photo-electron energies. From the graph in Figure 2, I roughly estimate that the SB minimum is

around 2.4 eV at the xMCQDPT2 level of theory, while the D0 state is (again rough estimate) at 3.5 eV. Thus, unless I made a mistake in reading these numbers, the energy gap is 0.9 eV. With a probe energy of 1.55 eV, the ejected electron would have a kinetic energy of 0.65 eV. However, in the text the authors write (line 154) that the energy is 0.87 eV. While the latter would indeed be very close to the experimental result of ~ 0.8 eV, the 0.65 estimate is a little farther off, possibly even beyond the expected error of the underlying theoretical method? Unless I made a mistake, I suggest the authors to check their numbers and, if needed, adjust the text and/or figure.

It is in fact very difficult to accurately determine where the peak maximum will be. In normal photoelectron spectroscopy, the difference between the energy minima (as suggested by the referee) will yield the adiabatic detachment energy. However, this does shift if we include the energy imparted into the many modes of the system. This means that one does not go from the bottom of one well to the same vertical point on the neutral. For this reason, we have taken essentially the “maximum” kinetic energy that we expect to see, rather than the peak which is difficult to determine. The peaks of the photoelectron spectra for all the intermediates is always shifted to the red of these maximal kinetic energies, which is what one would expect. We do not want to claim that the “static” calculations can accurately predict the spectrum and have opted instead for the approach we have used as a relatively fair consideration of the energetics. We have added some text to emphasise this.

Also, please include the structures, and energies as SI, as well as the CASSCF profiles. This would facilitate comparison to previous (and future) works.

We have done this.

The authors observe interesting oscillations in the photo-electron signals and attribute these oscillatory features to the wavepacket moving between the SB-twisted and the planar S1 minima. While this may be the case in vacuum, my understanding of oscillatory features in transient spectroscopy signals is that ultra-short pulses excite vibrationally coherent populations in the S1 state (and also S0, but that was not probed here, see e.g. Bardeen et al. Phys. Rev. Lett. 1995, 75, 3410). Thus the coherent vibrations would need to be Franck-Condon-active. Is the SB rotation coordinate also Franck-Condon active? Does the period of the (single) oscillation of the 1.4 eV signal match the vibronic progression?

The most likely FC modes involve C–C stretching because of the π^* excitation. The initial motion is fast and may be difficult to resolve in our experiment. But the wavepacket formed along the FC – S₁ min coordinate will then evolve into different motions (i.e. the isomerisation coordinate). The vibrational modes associated with this motion need not be FC active to become populated at later times. This is very similar to the dynamics of the GFP chromophore anion. Its dominant FC progression is ring scissor mode and imidazole ring breathing/in-plane CH bending mode, although it is well known from the time-resolved photoelectron spectroscopy that a twisted intermediate is also formed. We have added a discussion about this and the evolution from FC to S₁ min (i.e. the FC active wavepacket) and the subsequent evolution to the isomerisation coordinate.

The authors link their observations to the oscillations seen in the fluorescence upconversion experiments of Mataga and co-workers on the wtPYP protein (Chosrowjan et al. J. Phys. Chem. B 2004, 108, 2686). However, experimental and also computational evidence suggests that in the protein, only the DB-rotation pathway is active. If that is true, I do not see how the initial photo-

dynamics in PYP can be the same as in the isolated chromophore or that the protein would have no influence (line 213-216).

The reviewer is correct and it would be best to simply not make this comparison and avoid confusion. We have now removed this statement.

The authors mention that the signal at 0.8 eV decays with a timeconstant of 120 ps. Thus, I suppose the signal was measured for much longer time delays (at least up to 100s of ps) than displayed in figure 1. I suggest to include the complete data set as SI.

What is the origin of the slow 120 ps decay process?

We opted to focus on the early dynamics as this showed evolution of the angular distributions. We have included the long-time dynamics and spectra in the supporting information. The origin of the slow decay presumably is internal conversion to the ground electronic state, but because we cannot measure this directly, we have not commented on it.

Summarizing this is a very interesting and timely study that deserves to appear in Nature Communications as it would interest a broad community of chemists. However, the interpretation of the photoelectron spectra seems to depend strongly on the underlying computational model, which might have some accuracy issues. I am of the opinion therefore that the latter should be addressed before the manuscript can be published.

Reviewer #2 (Remarks to the Author):

General: this study reports interesting results from PAD measurements. This paper is written in a certain style using specific lab jargon. Nature Communication tends to the more general reader who might find these results not interesting, unless results and methodology are explained to an educated non-expert audience. The authors are strongly encouraged to do so, and do not leave it to the reader to consult the literature to put together the validity of the claims made here. Otherwise, this manuscript could be easily submitted to a more specialized journal.

It starts already with the photoelectron spectra (Figure S1). Would it not be possible for the authors to define VDE and ADE in the legend?

We have done so.

Would it not further be possible to spell out eKE?

We have done so now.

The main text mentions that the supplemental material provides details on photoelectron images ..., it does not. It does not even show a single detector image, not to mention a pipeline for data processing.

We have shown a representative image now and the analysis involved. Note that all raw data will be made freely available so that all images can be viewed.

The supplemental material reports that PAS were recorded with ns laser pulses. It is not clear why a photon energy of 3.45 eV avoids excitation of the excited state (to wit: "excitation is the least resonant with excited states of the anion"). Maybe the authors mean "photoproduct excitation".

No, we meant what we said. We have taken spectra at a number of photon energies and at 3.45 eV, there is the least signal from indirect processes. This therefore suggests that the chosen energy is a suitable energy to determine the VDE/ADE.

Further, along this line, the main text mentions 2.79 eV (444 nm) excitation with femtosecond pulses rather than using the 3.45 eV ns laser pulses as mentioned in the supplemental material. Would it not be possible to add one sentence to make this (trivial) distinction between the ns and fs laser pulses clear for the reader to avoid confusion?

We have done so now.

For pCA in PYP it is known that the ground and excited state cross sections are particularly favorable around 450 nm. This should be clearly stated and explained.

We have done so now.

Define Franck Condon (S0), SB(S1), DB(S1) and D0. A few words on LIIC pathway determination would be adequate.

These were defined appropriately in the manuscript “...two minima on the S_1 surface that arise from rotation around ϕ_{SB} or ϕ_{DB} with corresponding geometries SB and DB shown inset in Figure 2a”. S and D are standard text book representations of electronic excited states (singlet, doublet). We really do not see a need to define those any further nor the term Franck-Condon. The term LIIC is described in some more detail in the computational methods section. We feel that sufficient detail is provided, particularly given that these terms are pictorially defined in Figure 2.

Explain in plain words why the (aug)-cc-pVTZ basis affixed to the oxygens corrects for dynamic correlation (what dynamic correlation is relevant here), and why is this necessary.

The basis set doesn't account for dynamic correlation, the second order corrections arising from the method (XMCQDPT2) does. It is necessary if one wants to be able to obtain accurate vertical energies as it accounts for states coupling, which is particularly important as we are moving along a potential energy surface where states may become very close in energy. We have reworded to remove this confusion.

Explain in a few sentences “onion peeling”.

We have done so now.

More generally: The reader should have a chance to understand how lab and molecular frames can be connected without digging into the literature.

We believe we have: “Adiabatic changes can be tracked through molecular-frame PADs, but such measurements require a connection between the laboratory and molecular frames of reference, either through coincidence measurements or molecular alignment.” Also, we note that the point of the present study is that one does not need a connection to the molecular frame, so we feel that discussing this in detail distracts from the current work.

Discussion of Fig. 1 and Fig. 2 is overly complicated and full of lab jargon. For example, there is this sentence: Figure 2b shows Ψ_D for key critical geometries, YET Ψ_D is not mentioned anywhere in Figure 2b.

We have now labelled the Dyson orbitals as suggested. It is often difficult to use non-technical words as this takes away from the scientific accuracy. The Dyson orbital has been defined in non-lab jargon “The Dyson orbital can be thought of as the one-electron wavefunction describing the electron that is being photodetached.” It would not be appropriate to rename scientific terms that are used by a community for the sake of this paper as it would simply cause confusion for anyone willing to read the broader literature – so we do feel that some technical jargon is absolutely necessary.

Further there is the claim that lab frame PADs were calculated from the Ψ_D : Why not to show the calculated, time-resolved lab frame PADs and compare them with the observed PADs? Maybe they cannot be compared. If so, mention this.

We are a little confused by this statement as this comparison is shown in Figure 3. It is a central result of the paper to demonstrate the PADs are directly related to the electronic structure of the state being probed and that these can be quantitatively predicted using the relevant Dyson

orbitals.

The main conclusion of this paper is presented in a relative sentence following a main sentence that has nothing to do with single bond rotation: "...expected energy for the Franck Condon ..., while the broad peak ...". The authors are encouraged to learn how to present and highlight their main scientific findings to the non-expert. This publication should not be a lab manual.

The referee refers to the following sentence: "The agreement of the peak at $t = 25$ fs with the expected energy for the Franck-Condon (and S_1 planar minimum) geometry is excellent, while the broad peak centred at $\epsilon \sim 0.8$ eV is consistent with a twisted intermediate that has undergone rotation about ϕ_{SB} rather than ϕ_{DB} ." We have split this sentence up and have made it clear what the consequence of the observation is.

Fig. 2 is intriguing, but again, the discussion is strangely disconnected from the data. It is quite interesting to note that the mechanism in Fig. 2 might indeed coincide with ultrafast events observed in time-resolved crystallographic measurements on PYP (Pande et al., Science 2016). Maybe the authors comment on this.

It is important to appreciate that time resolved crystallography does not measure changes in *valence* electronic structure. Also, in the protein, the structural changes are influenced by the surrounding protein. We are not sure that such a comparison will offer much insight to the reader as the protein has an influence on the dynamics.

As non-Born-Oppenheimer dynamics is mentioned in the introduction, it requires some explanation why it appears here.

We have removed the reference to non-Born-Oppenheimer. It was there because many computational studies have looked to explore the conical intersections, but that's not relevant here.

For a successful publication in Nature communications, this reviewer recommends:

- avoid lab jargon
- argue for the laymen
- spell out the reasons why things were done, don't leave this to the imagination of the reader
- explain data processing and data analysis (in the supplemental material)
- in addition to pointing to literature, briefly explain the other methods employed in this paper.
- highlight and present main scientific findings separately, in main sentences, and in plain words. Avoid trying to explain two things in the same sentence.
- explain the data with findings from theoretical calculations by pointing out specific features. The more the reader understands the first time, the more valuable the results become.

We have tried to follow the referee's recommendation as much as we can and have made changes throughout the text to make things more digestible for a broad readership, while retaining the critical science that underpins the work.

Reviewer #3 (Remarks to the Author):

Review of manuscript by Prof Verlet and co-workers entitled "Geometric and electronic structure probed along the isomerisation coordinate of a photoactive yellow protein chromophore"

The present submission is one in a series by the Verlet group related to slightly different model chromophores of the Photoactive Yellow Protein (PYP). They include (as far as I have seen) in addition to the present submission:

[1] "Ultrafast valence to non-valence excited state dynamics in a common anionic chromophore". Nature Com. (2019)

[2] "Fingerprinting the Excited State Dynamich in Methyl Ester...." J. Phys. Chem A. Just accepted manuscript. (2020)

Although they address slightly different aspects of the photo physics of the PYP chromophore, the two former papers already contain parts (although not all) of information which is addressed in the present submission.

We strongly disagree with this conclusion. In the current manuscript, we show how photoelectron angular distributions provide a measure of the evolution of electronic structure along an adiabatic coordinate. In [1] we uncovered a new internal conversion (non-adiabatic) mechanism involving a dipole-bound state (not seen here) and [2] is essentially a follow-up of [1] with a focus on showing the consequences of such dynamics on the photoelectron spectra at low kinetic energy (not seen here). That the molecules are similar is immaterial to the scientific advance presented in this paper.

Why no reference to [1] is given in the present paper is unclear to me.

We had not referenced [1] because the focus of that study was internal conversion and dynamics of dipole-bound anions. We have included a short statement about these dynamics and how they are not seen in the present study despite the small structural differences.

General comments:

The paper is well-written and for the most part clear to follow.

The introduction of the present submission refers to the utilization of the time-resolved photoelectron imaging technique. In my mind, the application of this technique alone does not warrant a Nature Comm. paper. The combined electronic and nuclear dynamics is here indeed beautifully demonstrated by the time-resolved photoelectron imaging technique, but the technique is not brand new, and several reviews accounting on its various aspects exist (Stolow, Zanov, Toshinori, Neumark).

It is true that time-resolved photoelectron imaging is not new (and we never claim it to be). However, to the best of our knowledge, this is the first demonstration of lab-frame PADs providing a direct measure of electronic evolution along an isomerisation coordinate and the first time that such changes have been modelled accurately. All other cases have either focussed on dissociation or internal conversion, for which the changes in electronic structure are obvious. We have added a

statement at the end of the paper to emphasise this advance.

In the second paragraph, it is argued that the PYP protein is interesting and in particular the question about which specific bond isomerization occurs. It is true that the chromophore is particularly small and it is used as a model system for experiments along with theoretical studies. To the general readership of this Journal, this PYP model chromophore, however is perhaps not all that interesting. Other bio-chromophores for vision and energy harvesting might be more stimulating to read about.

Surely, this is in the eye of the beholder. PYP has been a key benchmark for many researchers world-wide, in particular for new advanced methods both experimental and theoretical. The present paper is part of this effort and despite the number of studies related to this chromophore, its precise photodynamics remains still elusive (as proven by the discussion with reviewer 1 above).

So, whether the system and the applied methodology is sufficiently new, groundbreaking, and interesting is to me questionable. That is not to say that the quality of the work is not high, I think it is!

Specific comments:

In figure 2a I need a discussion on why $\epsilon_{\text{(SB)}} = 0.87$ eV is used. If within, the time available between the pump and probe pulse, intramolecular vibrational redistribution (IVR) takes place, energy will be transferred to vibrations, which then is not available for the outgoing electron. Thus, the red arrow (upper left in figure 2a) could start on the S1 curve as well and hence give an $\epsilon_{\text{(SB)}}$ which is about 0.4 eV lower than 0.87 eV. The same goes for figure 2b. I think this needs a comment in the paper at least. To see if IVR has fully occurred, one could look at the kinetic-energy spectrum after for example 100 ps.

We have included a discussion of this (as also suggested by reviewer #1). We chose to retain the internal energy (i.e. "ignore" IVR) as we state that this will give the maximum energy. The maximum is expected at lower energy in agreement with our data. We have clarified this point by also explicitly stating the expected energy from the bottom of the potential. We agree that IVR will occur after 100 ps, but we really want to focus here on the first picosecond over which the initial isomerisation dynamics occur. The data over longer timescales are of poorer quality and do not actually show much spectral evolution.

The paragraph (line 192-196) discussing figure 2b did not provide me with intuitive insight. Could the authors perhaps elaborate on this?

We have now done so. The key point here is that we can intuitively understand the PADs based on the electronic structure. This is really quite powerful as PADs are often difficult to conceptually understand.

REVIEWERS' COMMENTS:

Reviewer #1 (Remarks to the Author):

The authors have addressed the issues brought up by the reviewers and revised their manuscript accordingly. For my part, all concerns have been dealt with adequately. In particular, the problem I had understanding the connection between the potential energy curves of Figure 2 and the kinetic energy of the electron no longer exists as it is now clearly explained.

During revision of their manuscript, a paper appeared in which single-bond rotation was also demonstrated experimentally for a PCK derivative, in which the double bond was blocked (Mushtalahti et al. Photoactive Yellow Protein Chromophore Photoisomerizes around a Single Bond if the Double Bond is Locked, *J. Phys. Chem. Lett.* 2020, 11, 2177). Although the results of this paper support one of the main conclusions in the current manuscript and can thus be considered relevant, the paper appeared after the first version of the present manuscript was submitted. I therefore want to leave it up to the authors to decide if they want to discuss the results of Mushtalahti et al. in the context of their own findings.

Reviewer #1 (Remarks to the Author):

The authors have addressed the issues brought up by the reviewers and revised their manuscript accordingly. For my part, all concerns have been dealt with adequately. In particular, the problem I had understanding the connection between the potential energy curves of Figure 2 and the kinetic energy of the electron no longer exists as it is now clearly explained.

During revision of their manuscript, a paper appeared in which single-bond rotation was also demonstrated experimentally for a PCK derivative, in which the double bond was blocked (Mushtalahti et al. Photoactive Yellow Protein Chromophore Photoisomerizes around a Single Bond if the Double Bond is Locked, *J. Phys. Chem. Lett.* 2020, 11, 2177). Although the results of this paper support one of the main conclusions in the current manuscript and can thus be considered relevant, the paper appeared after the first version of the present manuscript was submitted. I therefore want to leave it up to the authors to decide if they want to discuss the results of Mushtalahti et al. in the context of their own findings.

We have included a brief reference to this work. However, we refrain from making a detailed comparison on the grounds of the differing chemical structures and the focus here on gas-phase rather than solution- or protein-phase dynamics.